# What Is Missing in Food Loss and Waste Analyses? A Close Look at Fruit and Vegetable Wholesale Markets

**Ren Jie Zhang [1], Brian Lee [2] and Hung-Hao Chang [2],***

[1] School of Finance and Accounting, Fuzhou University of International Studies and Trade, Fuzhou 350202, China; zhangrenjie@fzfu.edu.cn

[2] Department of Agricultural Economics, National Taiwan University, Taipei 10617, Taiwan; r07627034@ntu.edu.tw

* Correspondence: hunghaochang@ntu.edu.tw; Tel.: +886-2-33662656

**Abstract:** Food loss and waste (FLW) has been the subject of significant research, with recent empirical evidence analyzing the determinants of FLW in many different countries. Much of this literature examines FLW at the farm, food processing, and consumer level. However, to the best of our knowledge, no study thus far has addressed this issue at food wholesale markets. This paper fills this knowledge gap by examining food loss at fruit and vegetable wholesale markets. Using a dataset with individual auction transactions from Taiwan, we examine the extent to which average daily market prices, product quality, and disaster shocks are associated with food loss. Results point to a negative effect of daily market prices on food loss. Furthermore, disaster shocks can lead to greater food loss, particularly typhoons. These effects are heterogeneous across the distribution for the weight of food loss.

**Keywords:** food loss and waste; wholesale market; fruit and vegetables; auction market; disaster shocks

## 1. Introduction

Policymakers and researchers have been highly concerned with food loss and waste (FLW) due to the increasing necessity for sustainable food systems that guarantee food production and consumption. Annually, roughly one-third of the food produced in the world (1.3 billion tons) is lost or wasted, costing $680 billion USD and $310 billion USD to developed and developing countries, respectively [1]. According to the Food and Agricultural Organization (FAO), food loss refers to decreases in the quantity or quality of food resulting from decisions or actions by food suppliers in the supply chain [2]. Food waste occurs when these losses are attributable to the decisions or actions by retailers, food producers, and consumers [2]. Since the production of food is resource-intensive, FLW is a salient issue, since it indirectly causes environmental issues such as soil erosion, deforestation, pollution, and greenhouse gas emissions [3]. In response, the United Nations declared the reduction of FLW by half as one of the key sustainable development goals to be achieved by 2030 [2].

Due to this growing attention from several stakeholders, a vast literature has examined the determinants of FLW in many countries. Several papers have identified an association between specific consumer characteristics and practices and food waste. For example, individual financial considerations are an essential factor in motivating the reduction of food waste [4–7]. Other studies suggest that a lack of knowledge between the negative ecological externalities caused by food waste leads to consumer indifference towards the issue [8,9]. Food-related household routines such as planning, shopping, storing, cooking, and managing leftovers can generate or reduce FLW as consumers prepare food at home [10–14].

Research has also examined the extent of food losses at different stages of the food supply chain. At the farm production level, food losses range from 20 to 40 percent for meat, dairy, cereal, and vegetable products [15–17]. These production losses occur when agricultural shocks such as weather conditions prevent crops from being harvested, or when produce is "graded out" based on quality [16]. Other literature has examined post-harvest losses of crops such as maize and sweet potato due to poor management skills and market mismatches [18,19].

However, despite this extensive research, no studies have been done on FLW at wholesale markets. There are two primary reasons to believe that FLW can be a severe problem at this stage in the food supply chain. First, many wholesale markets are unplanned urban spaces with no special facilities. Thus, these markets often lack even a single refrigeration unit for their products, particularly cold storage systems that can preserve and store goods for extended periods. Second, the auction system that typically governs wholesale markets can also generate FLW. Since consumers bid on both the quality and quantity of items, this means that products can be left unsold. Wholesale markets facilitate FLW because they have the unique cost of clearing their inventory of perishable products at the end of the business day. This clearance is done to ensure that there is adequate space for the following day's auction items.

This paper quantifies the determinants and extent of food loss for wholesale markets using a case study in Taiwan as an illustration. Using a large-scale dataset of individual transactions from a fruit and vegetable wholesale market, we estimate the weight of food loss using the ordinary least squares and quantile regression models. Results show that average market prices are negatively associated with the weight of fruit and vegetable food loss. Compared to high-quality grades of fresh fruits and vegetables, low-quality products are less likely to be sold within the auction market. Regarding the sourcing of agricultural products, we find that fresh fruits and vegetables provided by farmer's associations and agricultural production groups are less likely to be wasted in the wholesale market. Finally, we show that natural disasters, especially typhoons, affect FLW.

This paper makes several unique contributions to the burgeoning literature on food loss and waste. First, to the best of our knowledge, this paper is the only study that examines the determinants and extent of food loss at wholesale markets. This analysis is particularly significant since wholesale markets are touted as a solution to reducing FLW. For example, the FAO has suggested that wholesale markets minimize post-harvest food losses by establishing distribution pipelines that foster competitive markets while improving food-handling conditions [20]. However, the actual determinants and extent of FLW at wholesale markets are unknown. This contribution is especially relevant as governments continually support and integrate agricultural and wholesale markets, such as the Ethiopia Commodity Exchange and the *e*National Agricultural Market in India [21]. This paper provides the first estimates of food loss at wholesale markets using evidence from the largest fruit and vegetable wholesale market in Taiwan.

Second, we examine the heterogeneity of food loss in fruit and vegetable wholesale markets and their distributional determinants. The vast majority of papers examining FLW in the food supply chain exclusively consider one category of crops, such as vegetables, cereals, or maize [16–19]. We separately estimate the determinants of FLW for different types of vegetable products and fruits, respectively. This analysis allows us to precisely determine how food loss can differ solely based on crop type, since we examine fruits and vegetables within a single uniform market context. Additionally, we identify the determinants for small, medium, and large amounts of food loss. Much of the literature only studies the mean amount of food loss in the supply chain [16–19]. We leverage our rich dataset on individual fruit and vegetable transactions to determine how different amounts of food loss are attributable to varying factors while comparing between the magnitude of these effects. This distinction is crucial since it is unlikely that identical factors drive small and large amounts of food waste. In this study, we examine the distributional effect of food loss in the wholesale market by applying the quantile regression model.

Finally, we provide suggestive evidence on some of the short-term effects of climate change on food loss. An emerging literature has quantified the degree that global warming has intensified natural disasters in Taiwan, such as extreme rainfall and typhoons [22,23]. This study provides additional evidence on some of the mechanisms through which climate change can indirectly influence FLW since increased temperatures exacerbate natural disasters. This finding is especially relevant to other countries that are also susceptible to climate change and similar supply-side shocks.

The remainder of this paper is organized as follows. The next section provides background on wholesale markets for fruits and vegetables in Taiwan. We then follow with a description of the data. Next, we present the empirical framework and results. Finally, we conclude with a summary and discussion of policy implications.

## 2. The Wholesale Market for Fresh Fruit and Vegetables in Taiwan

The agricultural industry in Taiwan mainly consists of small-scale farmers that lack the capability to market and sell their own products. To address these concerns, the government legally established and supported farmer's associations, which organize marketing and joint production efforts among producers. Each county has a single farmer's association. These associations also complete other portions of the production process for farmers, such as coordinating the quality and quantity of production, distribution, grading and packing items, and market surveys. Notably, each farmers' association consists of several agricultural production groups. As of 2018, there are 302 Farmers' Associations consisting of approximately 5800 agricultural production groups that serve as local extensions of these organizations in Taiwan.

Wholesale markets operate by receiving their products from suppliers that purchase them from their local farmers. These markets collect and distribute agricultural products and serve as the primary channel for transactions between fruit and vegetable retailers and consolidators in Taiwan. In 2018, 1320 thousand metric tons of vegetable products were traded at food wholesale markets. Fruit and vegetable wholesale markets also employed a total of 1542 people at 52 different locations across the island [24]. Additionally, of the 349 million tons of fruits and vegetables marketed by agricultural cooperatives in 2018, 82% were shipped to the market in Taipei and handled by the Taiwan Agricultural Products Corporation (TAPC) [25]. Fruit and vegetable wholesale markets are also places where professional agents congregate to buy and sell products to other professionals in an auction-based system. Auctions are an efficient way of selling perishable products since they ensure that transaction prices represent the market equilibrium price. Therefore, each transacted price is equivalent to the market price at the time of the sale, meaning that wholesale markets are highly competitive.

The TAPC operates auctions at wholesale markets in the following manner. First, delivery trucks arrive at wholesale markets where goods are counted by tally clerks who then arrange the auction order. Second, the TAPC sends an evaluation team to grade the quality of each parcel (or box) of products. The initial prices of fruits and vegetables are determined based on the arrival volume, quality of products, climate, and festival/holiday demand. These prices are then forwarded to auctioneers. Finally, products are then sent to auction, which operates under the Dutch auction system. The auctioneer begins with the high initial price and lowers it until a participant accepts the price. Regardless of the initial price, auctions managed by the TAPC are recognized as completely open and transparent since a sale only requires one successful bid [26]. The TAPC operates the wholesale market and their auctions daily from 03:00 to 08:00, except for Sunday. Due to the facility's limited storage capability, the TAPC must clear all fresh food products by the end of the business day. Therefore, all fresh fruit or vegetable products must either be successfully sold in the auction market or sent to the disposal before the following day. Unsold parcels of fruit and vegetable products are labeled as food loss and are not returned to suppliers. At the end of each business day, the TAPC cooperates with the city government to dispose of these unsold products as garbage. This disposed weight of fruit and vegetable products measures the amount of food loss in wholesale markets and is the primary variable of interest.

## 3. Data

In this section, we introduce our datasets and the sample statistics of the selected variables.

### 3.1. Datasets

We use auction transactions from the fresh fruit and vegetable wholesale market sold at the Taipei Agricultural Products Corporation (TAPC), which is the largest wholesale market for fruits and vegetables in Taiwan. Our dataset contains the administrative records of all transactions sold under the auction system managed by the TAPC from 1 January to 31 December 2015. The total amount of fruit and vegetable products sold in the TAPC auction market during this year is approximately equivalent to 48.8 million kilograms. Of this quantity, 17,742 parcels of fresh fruits and vegetables were not sold in the auction market, or 2.03 million kilograms. Each parcel consists of a single type of fruit or vegetable since products are sorted and graded. For example, based on the categories established by the TAPC, vegetable products are further recognized into four groups—root and stem (e.g., radish, potato, and onion); leaf (e.g., cabbage, Chinese cabbage, and spinach); flower vegetables (cauliflower, cucumber, and winter melon); and others (e.g., mushroom). We do not further separate fruit products into subgroups because the TAPC provides no additional categorization for these products. These unsold parcels of fruit and vegetables are recorded as food loss by the TAPC. The weight of food loss at the TAPC is the dependent variable in this study.

Detailed information on each parcel of food is documented within the administrative profile of fruit and vegetable transactions at the TAPC. These characteristics include the date of arrival, final selling price, type of fruit or vegetable, the origin of suppliers, and product quality. For items that were not successfully sold in the auction, price is not available since they were disposed of at the end of the business day and classified as food loss. By law, these products cannot be resold to any consumer or institute nor donated to non-profits due to this classification. Similar to the specifications of previous studies examining FLW, we specify a continuous variable for the weight of each parcel of food loss in kilograms [11].

Several other variables are also used as controls. We sort transactions as either fresh fruit or vegetable sales. Since it has been documented that low-quality products are more likely to be lost in the FLW literature [27], we define several variables for product quality. In accordance with the information contained in the dataset, the quality of produce is categorized based on a grading system classifying products into excellent, good, normal, and low-grade produce. Finally, because different suppliers have varying standards and regulations in terms of quality and packaging, we specify dummy variables to indicate whether products were supplied by a farmer's association, agricultural production group, or individual farms, respectively. We calculate several variables for the daily average market price in NT $ per kilogram of fresh fruits and vegetables in each recorded transaction since market conditions may reflect both demand and supply-side factors and market prices are likely to be associated with food loss. Since natural disaster shocks in the production zones are highly associated with supply-side conditions, we identify whether the origin county of each parcel of food suffered from typhoons, heavy rainfalls, or other types of disasters in the previous two days before the current daily auction date. The Central Weather Bureau of Taiwan provided the data on natural disasters and weather conditions.

### 3.2. Sample Statistics

Table 1 reports the definitions and sample statistics of the selected variables. Out of the 17,742 parcels of fruit and vegetable products that were classified as food loss in the wholesale market, 3870 (22%) and 13,872 (78%) boxes consist of fresh fruit and vegetables, respectively. The average daily market price in the full sample is NT $37.01 per kilogram, while the price is higher for fresh fruits at NT $49.12 per kilogram. The majority of lost products is medium-quality produce. Concerning the sourcing of products, 52% and 35% of the fresh fruit and vegetable parcels were shipped to the TAPC by agricultural production groups and farmer's associations, respectively. Most of the fruits

were provided by farmer's associations, while the largest share of vegetable products were shipped by agricultural production groups. Summary statistics also show that typhoons affected food loss more than other types of natural disasters.

**Table 1.** Sample statistics.

| Variable | Definition | All Mean | All S.D | Fruit Mean | Fruit S.D | Vegetable Mean | Vegetable S.D |
|---|---|---|---|---|---|---|---|
| Weight | Weight in food loss (kg/parcel). | 114.58 | 250.13 | 100.95 | 223.03 | 118.38 | 257.06 |
| Market price | Daily average auction price of all fruit and vegetable products (NT $/kg). | 37.012 | 10.630 | 49.116 | 9.101 | 33.634 | 8.315 |
| Market price_fruit | Daily average auction price of fruit products (NT $/kg). | 49.287 | 10.072 | 49.096 | 9.095 | 49.341 | 10.328 |
| Market price_vegetable | Daily average auction price of vegetable products (NT $/kg). | 34.268 | 8.618 | 36.267 | 9.387 | 33.711 | 8.306 |
| Grade_excellent | If excellent grading (= 1). | 0.174 | 0.379 | 0.245 | 0.430 | 0.154 | 0.361 |
| Grade_good | If good grading (= 1). | 0.648 | 0.478 | 0.468 | 0.499 | 0.698 | 0.459 |
| Grade_normal | If normal grading (= 1). | 0.165 | 0.371 | 0.263 | 0.440 | 0.138 | 0.345 |
| Grade_bad | If bad grading (= 1). | 0.013 | 0.112 | 0.024 | 0.152 | 0.010 | 0.097 |
| Disaster_typhoon | If suffered from typhoon damage (= 1). | 0.024 | 0.152 | 0.036 | 0.187 | 0.020 | 0.141 |
| Disaster_rainfall | If suffered from heavy rainfall damage (= 1). | 0.019 | 0.136 | 0.002 | 0.048 | 0.016 | 0.126 |
| Disaster_other | If low temperature, strong wind or others (= 1). | 0.002 | 0.040 | 0.005 | 0.072 | 0.001 | 0.024 |
| Fruit | If fruit products (= 1). | 0.218 | 0.413 | 1.000 | 0.000 | 0.000 | 0.000 |
| Farmer association | If provided by farmer associations (= 1). | 0.351 | 0.477 | 0.494 | 0.500 | 0.311 | 0.463 |
| Agricultural production group | If provided by agricultural production groups (= 1). | 0.519 | 0.500 | 0.300 | 0.458 | 0.581 | 0.493 |
| Other supply | If provided by individual farms (= 1). | 0.130 | 0.480 | 0.206 | 0.405 | 0.109 | 0.311 |
| Number of parcel (N) | | 17,742 | | 3870 | | 13,872 | |

Note: Data were drawn from the Taipei Agricultural Products Corporation (TAPC) auction market of fruit and vegetable products.

## 4. Empirical Framework

The primary focus of this study is to estimate the determinants affecting the weight of fruit and vegetable parcels that are not sold in the auction market. We begin by specifying the following Equation:

$$ln(w_{ijt}) = \alpha + \beta_0 P_t + \beta_1 Dis1_{ijt} + \beta_2 Dis2_{ijt} + \beta_3 Dis3_{ijt} + \gamma' X_{ijt} + u_w + u_c + \varepsilon_{ijt}, \tag{1}$$

where $ln(w_{ijt})$ is the logarithm of the weight of each fruit or vegetable parcel classified as food loss shipped from county $j$ in date $t$. $P_t$ is the average daily auction price for fresh fruit and vegetable products in the TAPC market on day $t$. $Dis1$, $Dis2$, and $Dis3$ are indicator variables signifying if the origin county suffered from typhoon, heavy rainfall, or other types of natural disasters, respectively. We lag this variable using the previous two days of auction data to account for the potential transportation time lags from the production origin to the TAPC. $X_{ijt}$ is a vector of other explanatory variables associated with FLW including product grading, origins of production (see detailed list in Table 1), etc. $u_w$ and $u_c$ are week and county fixed-effects and $\varepsilon_{ijt}$ is the random error. There are 23 counties and 52 weeks. Since our dataset contains cross-sectional and time-series data, heteroskedasticity and

serial correlation may occur. We calculate the standard errors of the parameters using the Newey-West method to account for these two problems and to generate efficient estimators [28].

In our baseline model, we estimate Equation (1) using the ordinary least squares (OLS) method. Consistent estimates of parameters $\alpha, \beta_0, \beta_1, \beta_2, \beta_3, \gamma$ can be obtained using this specification. The OLS estimates capture the mean effect of the explanatory variables on food loss, although it fails to capture the distributional effects of the determinants on the outcome variable. To further investigate the distributional effects of the explanatory variables across the entire distribution of food loss, we estimate Equation (1) using the quantile regression (QR) method proposed by Koenker, R. [29]. Quantile regressions allow us to examine the potential heterogeneity of the explanatory variables' effects on the entire distribution of the dependent variable. In our case, using quantile regressions to estimate Equation (1) allows us to investigate whether the effects of auction market price, disaster shocks, and other factors vary depending on the amount of food loss.

Following Koenker, R. [29] and Koenker, R. and Bassett. Jr. G. [30], the conditional quantile regression of the weight of food loss corresponding to Equation (1) is rewritten as

$$ln(w_{ijt}) = \alpha_\theta + \beta_{0\theta}P_t + \beta_{1\theta}Dis1_{ijt} + \beta_{2\theta}Dis2_{ijt} + \beta_{3\theta}Dis3_{ijt} + \gamma_\theta'X_{ijt} + u_w + u_c + \varepsilon_{ijt\theta} \qquad (2)$$

where $\theta$ indicates the quantile of food loss conditioned on the exogeneous vectors. The distribution of the error term $\varepsilon_{ijt\theta}$ is left unspecified. Thus, the only identifying conditional requirement for Equation (2) is that the conditional quantile evaluated at each quantile is zero. Accordingly, the coefficients $\alpha_\theta, \beta_{0\theta}, \beta_{1\theta}, \beta_{2\theta}, \beta_{3\theta}, \gamma_\theta$ capture the effects of auction market price, disaster shocks, and other determinants at the $\theta_{th}$ percentile of the food loss distribution, respectively. Equation (2) is estimated using the generalized method of moments (GMM) framework [29]. Although the standard errors of the parameters can be derived from the GMM estimation, these standard errors severely understate the standard deviations of the estimators [31]. To increase the efficiency of the estimators, the standard errors of the quantile regression estimates are obtained using a bootstrapping method with 500 replications, which have been shown to perform better than the asymptotic standard errors derived from the GMM [32].

## 5. Results and Discussion

### 5.1. Main Findings

Column A of Table 2 presents the estimation results of the OLS model for all food loss. For the model's identification purpose, the baseline comparison category is low-quality products. Overall, food loss at wholesale markets in Taipei is about 4%. Additionally, market prices are associated with reduced amounts of food loss, as a one dollar increase in the average daily auction price of fruits and vegetables lowers the weight of food loss by 0.4%. This result is expected as higher average market prices may reflect stronger consumer demand on that particular day. Consumers with limited budgets are more likely to purchase relatively low-quality products when the market prices are high. This finding reinforces prior research showing that market conditions are one of the most significant factors in generating FLW [18,19]. Additionally, parcels of food loss that are classified as excellent or high-quality are 70.8% and 50.3% lighter than low-quality parcels, *ceteris paribus*. This result is in line with research demonstrating that consumers are willing to pay for superior produce, which is an essential component in the demand for fresh food [33].

As expected, natural disaster shocks are correlated with increased levels of food loss. If a county suffered from typhoons during the previous two days, then the weight of food loss for parcels of fruits and vegetables sourced from these locations increased by 34%, respectively. Unsurprisingly, exogenous shocks increase food loss since they destroy agricultural assets and infrastructure while causing production losses [34]. Food loss from fruits and vegetables provided by farmer's associations and agricultural production groups is 52.9% and 37.7% lighter, *ceteris paribus*. One explanation for these findings is that farmer's associations and agricultural production groups are directly and indirectly

regulated by the Council of Agriculture (COA), respectively. Thus, produce sourced from farmer's associations comply with stricter quality controls as a result of this government oversight. Additionally, both of these organizations have internal inspections and processes to ensure the quality of produce.

**Table 2.** Estimation of the weight equation for food loss.

| Variable | Column A All | | | Column B Fruit | | | Column C Vegetable | | |
|---|---|---|---|---|---|---|---|---|---|
| | Coefficient | | S.E | Coefficient | | S.E | Coefficient | | S.E |
| Market price_all | −0.004 | *** | 0.001 | | | | | | |
| Market price_fruit | | | | −0.003 | | 0.003 | | | |
| Market price_vegetable | | | | | | | −0.003 | ** | 0.002 |
| Grade_excellent | −0.708 | *** | 0.099 | −0.292 | ** | 0.142 | −1.028 | *** | 0.130 |
| Grade_good | −0.503 | *** | 0.098 | 0.027 | | 0.140 | −0.902 | *** | 0.129 |
| Grade_normal | −0.301 | *** | 0.099 | 0.053 | | 0.142 | −0.662 | *** | 0.131 |
| Disaster_typhoon | 0.340 | *** | 0.058 | 0.130 | | 0.098 | 0.389 | *** | 0.072 |
| Disaster_rainfall | 0.018 | | 0.060 | −0.063 | | 0.121 | 0.111 | * | 0.068 |
| Disaster_others | 0.132 | | 0.333 | 0.292 | | 0.458 | −0.198 | | 0.411 |
| Fruit | 0.024 | | 0.029 | | | | | | |
| Farmer association | −0.529 | *** | 0.044 | −0.813 | *** | 0.069 | −0.329 | *** | 0.057 |
| Agricultural production group | −0.377 | *** | 0.044 | −0.718 | *** | 0.073 | −0.208 | *** | 0.056 |
| Constant | 4.908 | *** | 0.135 | 4.425 | *** | 0.262 | 5.216 | *** | 0.172 |
| Week fixed effects | | Yes | | | Yes | | | Yes | |
| County fixed effects | | Yes | | | Yes | | | Yes | |
| Adjusted $R^2$ | | 0.130 | | | 0.208 | | | 0.133 | |
| N | | 17,742 | | | 3870 | | | 13,872 | |

Note: The dependent variable is the logarithm of weight. The Newey-West standard errors are reported. ***, **, * indicates significance at the 1%, 5%, and 10% level.

### 5.2. Results of the Heterogeneity Analysis Separating for Fruits and Different Types of Vegetables

Column B of Table 2 reports the results of the heterogeneity analysis for fruits. In contrast to the OLS model for all food loss, market prices have no significant impact on the weight of food loss for fruits. This lack of significance is likely attributable to the fact that price is no longer the only indicator of product quality due to changing consumer preferences. For example, research has shown that consumers in Taiwan are willing to pay a premium for safe fruits, which is an especially important concern since they are either consumed raw or juiced [35]. Fruit loss classified into the excellent-quality category is 29.2% lighter than low-quality fruit. However, fruits classified as medium-quality are no longer correlated with reduced food loss, suggesting that consumers have a stronger demand for better fruit. Medium-grade fruits tend to accrue minor cosmetic blemishes when they are being transported, resulting in lower consumer demand since fruits are consumed without further processing [36]. The sourcing of fruits is especially important since fruit loss from farmer's associations and agricultural production groups is 81.3% and 71.8% lighter than parcels provided by individual farmers. Table 3 displays the findings of the heterogeneity analysis by different types of vegetables based on their TAPC classification, including root and stem, leaf, and flower vegetables, respectively. We calculate separate market prices for these three categories of vegetables and find that increases in the price of leaf and flower vegetables decrease the weight of food loss by 0.4% and 0.3%, respectively. Meanwhile, food loss from root and stem vegetables graded as excellent or good is significantly lighter in comparison to leaf and flower vegetables. One explanation for these findings is that these categories of vegetables have different lengths of perishability. For example, consumers are less likely to be sensitive to market prices for root and stem vegetables since they have an extended post-harvest life during storage [37]. Thus, when consumers buy root and stem vegetables, they are primarily driven by quality, as they do not need to be purchased daily for freshness. In contrast, leaf and flower vegetables have a short post-harvest life, and consumers purchase them more frequently to guarantee quality.

**Table 3.** Estimation of the weight equation of food loss by type of vegetables.

| Variable | Model A (Root and Stem Vegetables) Coefficient | | S.E | Model B (Leaf Vegetables) Coefficient | | S.E | Model C (Flower Vegetables) Coefficient | | S.E |
|---|---|---|---|---|---|---|---|---|---|
| Market price_by type [#1] | 0.002 | | 0.005 | −0.004 | ** | 0.002 | −0.003 | ** | 0.001 |
| Grade_excellent | −2.386 | *** | 0.584 | −1.610 | *** | 0.160 | −0.957 | *** | 0.271 |
| Grade_good | −2.127 | *** | 0.583 | −1.349 | *** | 0.151 | −1.021 | *** | 0.270 |
| Grade_normal | −1.656 | *** | 0.593 | −1.107 | *** | 0.153 | −0.963 | *** | 0.274 |
| Disaster_typhoon | −0.134 | | 0.172 | 0.855 | *** | 0.146 | 0.380 | *** | 0.081 |
| Disaster_rainfall | −0.302 | ** | 0.154 | 0.524 | *** | 0.114 | 0.229 | *** | 0.088 |
| Disaster_others | 0.005 | | 0.035 | 0.024 | | 0.231 | −0.082 | | 0.362 |
| Farmer association | −0.310 | *** | 0.094 | −0.437 | *** | 0.125 | −0.284 | *** | 0.096 |
| Agricultural production group | −0.256 | *** | 0.095 | −0.456 | *** | 0.123 | −0.208 | ** | 0.097 |
| Constant | 6.096 | *** | 0.697 | 5.766 | *** | 0.264 | 4.950 | *** | 0.340 |
| Week fixed effects | Yes | | | Yes | | | Yes | | |
| County fixed effects | Yes | | | Yes | | | Yes | | |
| Adjusted $R^2$ | 0.128 | | | 0.148 | | | 0.074 | | |
| N | 2,016 | | | 6,979 | | | 4,659 | | |

Note: #1 market price is calculated corresponding to each category of vegetables in the regression model. The dependent variable is the logarithm of weight. The Newey-West standard errors are reported. ***, ** indicates significance at the 1% and 5%, level.

Food loss from root and stem, leaf, and flower vegetables considered to be of normal-quality is also lighter than their low-quality counterparts. This result is attributable to the fact that consumers are more willing to purchase standard-quality vegetables since these products tend to go towards food service where they are then cut, making appearances less significant [36]. Natural disaster shocks are still significant, as leaf and flower vegetable losses originating from counties that experienced typhoons or intense rainfall is heavier by 22.9–85.5%, respectively. Finally, vegetable losses sourced from farmer's associations and agricultural production groups is 20.8–45.6% lighter than parcels supplied by individual farms.

### 5.3. Results of the Distributional Effects

Table 4 reports the results of the quantile regression model examining the effects of the explanatory variables on the distribution of food loss. Our results indicate that several factors affect small, medium, and large amounts of food loss at the 0.25, 0.50, and 0.75 quantiles of the weight distribution.

**Table 4.** Estimation results of the quantile regression model.

| Variable | QR25 Coefficient | | S.E | QR50 Coefficient | | S.E | QR75 Coefficient | | S.E |
|---|---|---|---|---|---|---|---|---|---|
| Market price_all | −0.004 | *** | 0.001 | −0.006 | *** | 0.002 | −0.003 | ** | 0.002 |
| Grade_excellent | −0.481 | *** | 0.091 | −0.413 | *** | 0.095 | −0.930 | *** | 0.102 |
| Grade_good | −0.301 | *** | 0.089 | −0.234 | ** | 0.093 | −0.708 | *** | 0.100 |
| Grade_normal | −0.188 | ** | 0.091 | −0.062 | | 0.095 | −0.519 | *** | 0.101 |
| Disaster_typhoon | 0.324 | *** | 0.065 | 0.287 | *** | 0.068 | 0.333 | *** | 0.072 |
| Disaster_rainfall | 0.105 | | 0.074 | 0.094 | | 0.077 | 0.046 | | 0.083 |
| Disaster_others | −0.393 | | 0.298 | −0.158 | | 0.311 | 0.756 | ** | 0.333 |
| Fruit | 0.034 | | 0.033 | 0.126 | *** | 0.035 | 0.079 | ** | 0.037 |
| Farmer association | −0.247 | *** | 0.045 | −0.480 | *** | 0.047 | −0.668 | *** | 0.050 |
| Agricultural production group | −0.097 | ** | 0.045 | −0.368 | *** | 0.047 | −0.626 | *** | 0.050 |
| Constant | 3.842 | *** | 0.134 | 4.639 | *** | 0.140 | 5.878 | *** | 0.150 |
| Week fixed effects | Yes | | | Yes | | | Yes | | |
| County fixed effects | Yes | | | Yes | | | Yes | | |
| Pseudo $R^2$ | 0.092 | | | 0.085 | | | 0.075 | | |
| N | 17,742 | | | 17,742 | | | 17,742 | | |

Note: The dependent variable is the logarithm of weight. Standard errors are calculated using the bootstrapping method with 500 replications. ***, ** indicates significance at the 1% and 5% level.

Across all quantiles of the distribution, increases in the daily average auction price of fruits and vegetables reduces the weight of food loss by 0.3—0.6%, *ceteris paribus*. Figure 1 depicts the distributional effects of auction prices on the weight of food loss. These results suggest that higher demand for fruits and vegetables has more significant effects on larger amounts of food loss. Food loss categorized as excellent-quality is 41.3—93% lighter than low-quality parcels, with these effects becoming more pronounced at the upper ends of the distribution. Similar findings are also observed for medium-quality produce, although the magnitude of these effects is not as large.

Natural disasters are also significant across quantiles of the food loss distribution. For example, typhoons increase the weight of food loss by 32.4%, 28.7%, and 33.3% at the 0.25, 0.50, and 0.75 quantiles, respectively. Figure 2 shows the distributional effects of disasters on the weight of FLW. Natural disaster shocks impact food loss since they are supply-side shocks that affect the production of fruits and vegetables in Southern Taiwan. Thus, events like typhoons impact the quality and quantity of fruits and vegetables before their shipment to Taipei, located in Northern Taiwan. Food loss sourced from farmer's associations is 24.7%, 48%, and 66.8% lighter than loss provided by individual farmers at the 0.25, 0.50, and 0.75 quantiles. Similarly, food loss sourced from agricultural production groups is 9.7%, 36.8%, and 62.6% lighter than loss provided by individual farmers at each respective quantile.

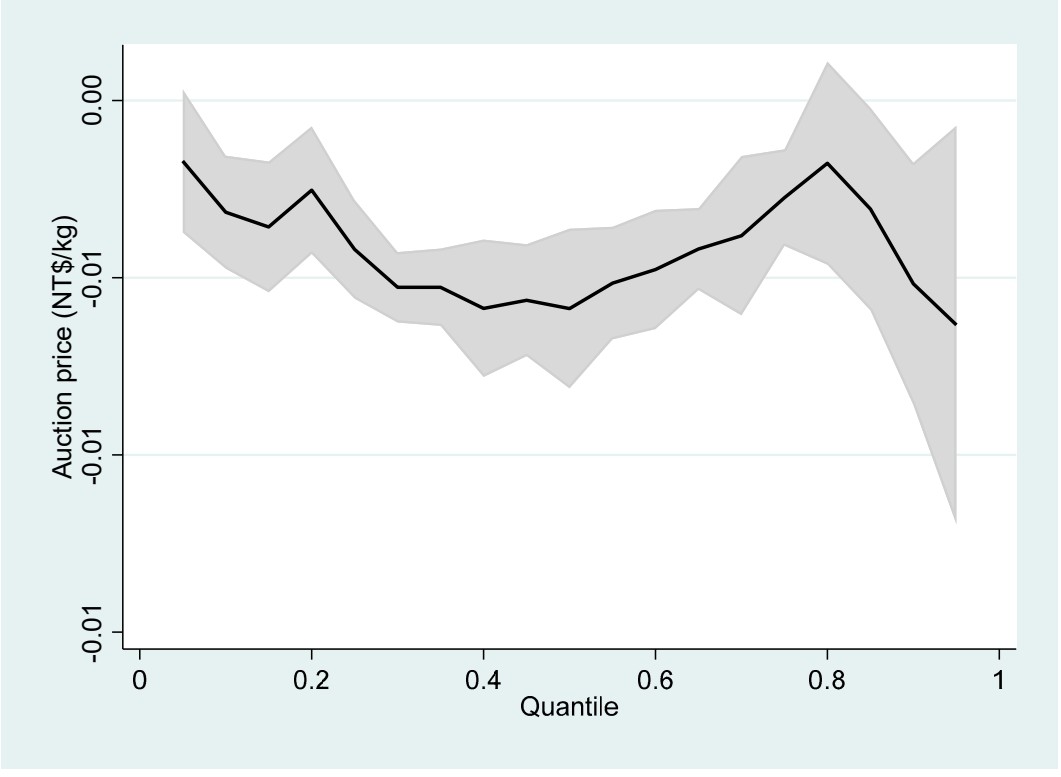

**Figure 1.** The distributional effects of auction price on food loss in weight. Note: Results are summarized from the quantile regression model. The solid line is the estimated effect of market price on the weight of food loss at each parcel. The shadowed area is the 95% confidence intervals.

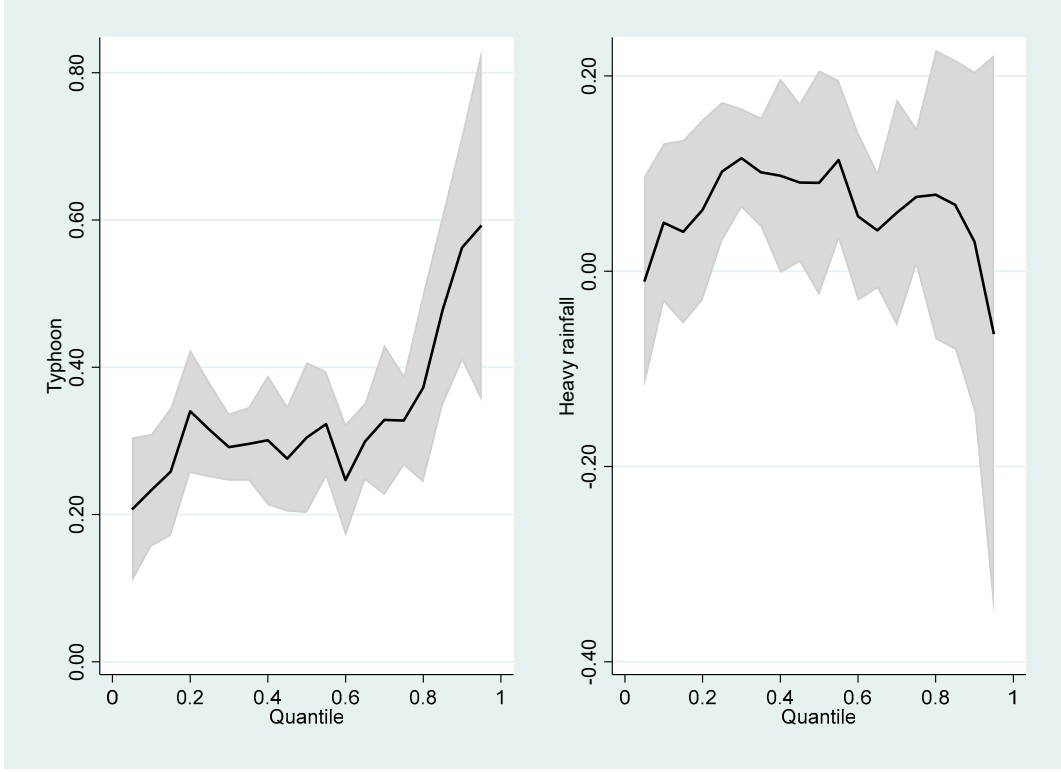

**Figure 2.** The distributional effects of disasters on food loss in weight. Note: Results are summarized from the quantile regression model. The solid line is the estimated effect of the disaster shocks on the weight of food loss at each parcel. The shadowed area is the 95% confidence intervals.

*5.4. Discussion*

The findings of this paper explain several patterns related to FLW in food wholesale markets. First, we show that food loss in food wholesale markets is smaller when compared to other stages of the food supply chain and food waste generated by consumers. In terms of pure weight, the amount of food loss at fruit and vegetable wholesale markets in Taiwan is approximately 4%. To compare, Buzby, J. et al. [38] calculated the total amount of food loss and waste for the US in 2010. The authors find that retail and consumer-level food loss was 10% and 21% of the available food supply, respectively. Our estimates suggest that wholesale markets are efficient mediums for selling produce to consumers. For example, issues such as dented cans and damaged packaging are likely to be smaller concerns at fruit and vegetable wholesale markets since retail food providers also purchase their ingredients from these locations. However, future studies could also examine the feasibility of using an alternative auction structure such as an English auction to ensure that products receive an initial bid. Second, FLW in food wholesale markets can be further reduced by using advanced smart agricultural technologies such as cold chain storage. While the amount of FLW at food wholesale markets is small, fruits and vegetables also pass through other intermediaries before arriving at markets like the TAPC. Thus, reducing FLW at wholesale markets remains an essential step in mitigating food loss during the supply chain. Prior literature such as Minten, B. et al. [39] shows that infrastructure improvements such as cold chain storage are associated with increased efficiency in food value chains by reducing food loss. An implication of our findings is that the use of additional storage and handling technologies can allow wholesale markets to keep food for extended periods, allowing for the maximization of agricultural efficiency.

Third, while natural disasters have direct consequences on agricultural production in farm production fields, we show that they can also indirectly affect food loss at food wholesale markets. This loss is attributable to the fact that agricultural shocks disrupt the daily transportation networks used

to move products to locations such as wholesale markets [34]. These interruptions are particularly relevant for auction markets since products are sourced daily to ensure freshness and rotation. Therefore, wholesalers and local governments should work in tandem to find alternative uses for fruits and vegetables that cannot be transported to wholesale markets because of natural disaster shocks. An example of such an initiative could include donating produce to local schools.

Finally, our results suggest that governments can implement policies to reduce FLW at food wholesale markets. For example, local authorities in Taiwan are already involved in measuring and transporting food loss to the disposal. Rather than throwing away these edible products, wholesale markets and officials should implement policies where unsold products are donated to non-profits such as charities or food banks. One successful example includes France, which banned supermarkets from throwing away edible food in 2016. This law increased the quantity of donations available at food banks, with 5000 of these charities now receiving nearly half of their donations from supermarkets [40]. However, an important consideration for these programs is ensuring that the quality of donated products remains high since many of these products tend to be unsold or of lower-quality. Similarly, other policies that could be implemented include those used in Japan's Food Waste Recycling Law, enacted in 2001 [41]. These laws target food waste generated by food suppliers and require them to recycle as appropriate. Our findings suggest that other retailers, such as wholesale markets, could also minimize their FLW by adopting similar practices.

## 6. Conclusions and Research Limitations

This study examines food loss at fruit and vegetable wholesale markets in Taiwan. While there is a considerable body of literature on this topic, our study makes several unique contributions. First, we quantify the determinants and extent of food loss at wholesale markets. Second, we analyze the determinants of food loss separately for fruits and different types of vegetables and across the entire distribution of food loss. Finally, our results provide suggestive evidence on the indirect effects of climate change on FLW through natural disasters. We find that daily market prices and higher-quality fruits and vegetables are negatively associated with food loss. Furthermore, natural disasters such as typhoons can generate food loss. All of these effects are heterogeneous across the distribution of food loss.

There is still much work to be done to understand the determinants and extent of FLW in the food supply chain. First, one limitation of this study includes the fact that we are only able to include fruits or vegetables in our sample. Research examining FLW at food wholesale markets for other commodities such as meat or dairy would be insightful. Second, our measures of product quality calls for caution. In accordance with the information contained in the dataset, we are only able to use the objective measures of food product quality based on the expertise of the TAPC. Future studies can check the robustness of our findings using an alternative measure of the product quality. Third, food loss at wholesale markets may also depend on consumer demand. If detailed information on consumers becomes available, then we could better control for these factors in our regression analysis. Fourth, our estimates may suffer from omitting variable bias since the transportation costs of each parcel is unavailable. Finally, this paper only evaluates food loss at one food wholesale market in Taiwan that is based on the auction system. Additional scholarly work could study FLW in other countries or markets in order to further examine the relationship between different selling systems and food loss.

**Author Contributions:** Each author has equal contribution to the manuscript in term of data collection, model estimation and writing.

**Funding:** This research received no external funding.

**Acknowledgments:** Hung-Hao Chang's time on this paper is supported by the research project #108-2410-H-002-053 of the Ministry of Science and Technology. Brian Lee acknowledges the sponsorship of Fulbright Taiwan (Foundation for Scholarly Exchange).

**Conflicts of Interest:** The authors declare no conflict of interest.

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
