# Peer review of "What Is Missing in Food Loss and Waste Analyses? A Close Look at Fruit and Vegetable Wholesale Markets"

_sustainability, doi:10.3390/su11247146_

Round 1
Reviewer 1 Report
Comments and suggestions are enclosed in the file below.

Reviewer 2 Report
Summary
This manuscript studies food loss and waste (FLW) at a fruit and vegetable wholesale market in Taiwan. Using a rich dataset from the largest wholesale market in Taiwan, they examine the determinants of daily FLW. They study the way average daily auction prices, product quality, disaster shocks, and origin of food affect the percentage of FLW. They employ two econometric strategies: Ordinary Least Squares and Quantile Regression to answer their research question.
Broad Comments:
The manuscript is very interesting as fills a gap in the literature. The authors have access to a rich dataset that help them answer the research questions. The authors should explain the difference between food loss and food waste? From their estimation strategy, they don’t seem different. If this is a term widely used in the literature, it is worth defining it and explaining its origin. The authors can exploit the rich dataset better, using panel data regression and including time (daily or weekday) effects and county fixed effects. At the same time, they should focus on particular fruits or vegetables (See specific comments) In general, the authors should explain the intuition behind the results. Why do the coefficients have the signs they have? For example, why are average daily auction price and FLW positively correlated? Why do natural disasters increase FLW? The authors should acknowledge more limitations of their methodology and data.
Specific Comments:
Abstract:
[Lines 17-19]: The authors also use a quantile regression. While I don’t necessarily think both OLS and quantile regressions are needed in the abstract, if OLS is included, quantile regression should also be included.Introduction:
[Lines 88-90] Isn’t the aggregation worst for estimation purposes? If the authors have detailed data on each type of fruit and vegetable, they should precisely estimate the FLW for each type of fruit or vegetable as there may be systematic differences across each type. For example, a particular fruit/vegetable may be more perishable than another. Similarly, a particular fruit may be more popular for juices as opposed to direct consumption. These idiosyncratic differences make the aggregate analysis less precise.The Wholesale Market for Fresh Fruit and Vegetables in Taiwan.
[Lines 124-125]: What is the share of fruit and vegetables that are traded in food wholesale markets? Provide a detailed description of the auction system for context. [Line 139]: Does each parcel contain a specific fruit/vegetable or a combination of different products? Detailed description about the parcels is needed (size, content, etc) [Lines 141-143]: If the weight of the fruit and vegetable products measures the amount of food loss and waste, what is the difference between food loss and food waste? As stated before, why is this term used as opposed to food loss or food waste?Datasets:
When is the data from? Provide range of dates. Was it for the entire year of 2015? It is not clear. [Lines 151-153]: What is the percentage of FLW? State the number of observations upfront Please clarify whether the unsold fruits and vegetables are all disposed as waste at the end of the day. In other words, is there any possibility in which the fruit is kept for more than one day or returned to the cooperative/supplier if unsold? [Line 164]: Why not do the analysis for each specific food instead of categorizing them as fruit or vegetable? [Line 168]: which other sources? What other information? [Lines 169-170]: How was the average auction price calculated? Does the dataset have auction prices for each type of fruit/vegetable each day? Using the average auction price is a major limitation as the auction price of the particular fruit or vegetable should influence the food loss for that specific food.Sample Statistics
What time period does the dataset cover? For the quality measure, is there any measurement error? The three categories seem very broad.
Empirical Analysis
Since the authors have information on the location where the food come from, they should use the distance travelled and county fixed effects as controls in their estimation. Equation (1) has notational and specification problems (same comments apply to Equation (2). The monthly fixed effects should exclude a month as the model includes an intercept. What does m stand for? The time fixed effects should have coefficients, even if these are not reported. Why are the authors using monthly fixed effects instead of weekday/daily fixed effects? If the data is analyzed at the individual food level, the daily fixed effects would be more appropriate. Do you have multiple observations per day? Define ‘ and avoid using x for multiplication. Why doesn’t the first Beta have a subscript? What does X include? Define k For the Dis variable, why are the authors aggregating all three disasters? The authors should include a dummy variable for each type of disaster. Why are the authors using a 2-day lagged? How is that number determined? For certain crops, aren’t longer lagged more appropriate. The model does not include county fixed effects. The model should control for the origin (ideally distance traveled) of the products in the parcel. The authors should not aggregate the data by vegetable, fruits, or both. They should also employ the auction price for the particular type of food (instead of the daily average) Lines [192-197]: use mathematical notation and subscripts to match equation. Lines [199-201]: the authors should argue why the assumptions behind OLS hold such that the estimates are consistent The authors should be careful as the assumptions don’t hold given the aggregation, heteroscedasticity, possible multicollinearity between price and quality, and omitted variable bias. Line [202]: Don’t the authors use panel data? There is a time element in Equation (1). Line [205]: What’s the difference between quantile and conditional quantile regression?Results and Discussion
Since the estimation results will have to be updated given the comments above, I would only comment on some of the intuition that needs clarification in this section. Once the manuscript is revised, I will comment more on the results and conclusions. The authors interpret average daily auction price effects as signaling quality. However, the models already control for some quality. What’s the intuition behind the price effect? What’s the intuition behind natural disaster shocks and FLW? Since the authors combine the three possible disasters in one dummy variable, this interpretation is harder. As stated in the previous section, the authors should control for each shock separately and explain the intuition behind the results. Shouldn’t natural disasters such as typhoons also affect the demand for food?Research Limitations:
What are the limitations of the data? What are the limitations of the econometric model?
Table 1:
Authors may include statistically significant symbols (i.e. *, **, ***) to show the results of two-sample t-tests between fruit and vegetable datasets.Table 2:
Are the authors reporting robust standard errors? Are the authors reporting adjusted R^2s? Moreover, the authors should comment about the low R^2 levels in the models.Table 3:
Are the authors reporting robust standard errors? Why report the 10th and 90th quantiles instead of the .25 and .75?Figure 2:
Panel B shows a large confidence interval for the last quantile, which includes zero. The authors should comment about this.The authors do not state the division of the responsibilities for the manuscript as required by the journal.
Reviewer 3 Report
Very good work!
Author Response
Dear reviewer,
Thank you very much for your kind words on our paper.
Round 2
Reviewer 2 Report
Summary
While the manuscript improved, there are still a number of issues that should be addressed before publication.
Broad Comments:
In general, the authors should better explain the intuition behind the results. The authors should acknowledge more limitations of their methodology and data. While more information about the auction was provided, it lacks a connection to auction theory. The authors appear to describe a Dutch auction. They should explain the optimal/theoretical strategy in such auction and how the reserve price is established. For example, what if 2 people accept a price? I recommend including more context and references from auction theory.
Specific Comments:
Introduction:
[Lines 34-35] Cite source. [Lines 56-58]: Is that statement unique of auctions? There are other instances in which products are left unsold. [Lines 81-83]: saying that this papers examines the determinants and extent of food loss is a statement that implies causality. However, the models employed and the data limitations do not allow for causality. If a causality case is going to be made, the authors should explain how the methods are valid for such causal claims. [Lines 96-98]: What is the intuition behind the price and food loss relation?The Wholesale Market for Fresh Fruit and Vegetables in Taiwan.
[Lines 136-144]: as stated in the broad comments, a connection to auction theory is very important. [Lines 136-144]: Does the evaluation team determine the initial prices or is it the suppliers? Why aren’t these initial prices used in the regression analysis? I assume these initial prices are available to the authors and may explain why the auction wasn’t successful. The authors should employ this information in the regression analysis. Moreover, if the prices are available per parcel, a daily fixed effect may be possible.Data:
[Line 162-164]: While the additional information improves the article, the authors should compute the percentage of food loss, which they refer to in the discussion section [Lines 343-344].They should present this information before the end of the manuscript. The authors claim to have added this information on Lines 162-165. However, that information is not there. [Lines 175-176]: why aren’t the authors using the initial price per parcel? [Lines 169-170]: How was the average auction price calculated? Does the dataset have auction prices for each type of fruit/vegetable each day? Authors claim to provide more details in lines 188-191. However, it is still not clear how the raw data looks like. How many observations are used to compute the daily average? Do you have multiple observations per day? It is very important if the authors are clearer about the raw data and its content. What variables are observed? How many observations are available per day?Sample Statistics
For the quality measure, is there any measurement error? The three categories seem very broad.
Empirical Analysis
Equation (1) has notational and specification problems (same comments apply to Equation (2). The week and county fixed effects need a better description. How many counties are there? Are the authors using week or weekday fixed effects? The latter may be more appropriate. For example, there may be differences between Monday or Saturday markets. The authors should try including such fixed effects and see if the fit is better. Equation (1) should exclude an intercept given the fixed effects. The time and county fixed effects should have coefficients, even if these are not reported. The authors must define Why doesn’t the first Beta have a subscript? What does X include? Instead of listing etc [Line 221], the authors should list all the variables used in the model. If these variables are all summarized in a table, the authors should refer to that table. For the Dis variable, why are the authors aggregating all three disasters? The authors should include a dummy variable for each type of disaster. While the authors say that there is only enough data for rainfall or typhoons, the authors should separate these two disasters. From the regression tables, the aggregation does not seem as clear. Is there a dummy variable (=1) if a typhoon took place 2 days before? The authors should clarify how these natural disasters enter the regression. Why are the authors using a 2-day lagged? How is that number determined? For certain crops, aren’t longer lagged more appropriate. Does all transportation take place within 2 days? The authors didn’t address this issue fully.The authors should be careful as the assumptions don’t hold given the aggregation, heteroscedasticity, possible multicollinearity between price and quality, and omitted variable bias. While the authors claim to have addressed this issue, they still make causal claims that rely on OLS assumptions. [Line 227]: the authors claim that they have consistent estimates of the parameters in their model. However, this consistency relies on assumption that don’t hold in this model. For example, the model suffers from omitted variable bias as the authors don’t observe daily prices per parcel. The authors should state which model is more reliable among the different regressions that were estimated. Why did the authors not categorize fruits as they did for the vegetables?
Results and Discussion
[Lines 258-260]: Shouldn’t the intuition behind the price results be connected to the Dutch auction system? How do you interpret higher bids in a Dutch auction? What is the intuition behind this? [Lines 273]: lighter than… [Lines 286-288]: the interpretation of the coefficient not being statistically significant is incorrect. They interpretation of this coefficient is relative to the low-grade category. [Lines 318-319]: How are the quantiles chosen? Does the literature provide guidance? Since the estimation results will have to be updated given the comments above, I would only comment on some of the intuition that needs clarification in this section. Once the manuscript is revised, I will comment more on the results and conclusions. Discussion: the authors should connect food loss with the auction system. The authors should also discuss the implications of findings connected to farmers’ associations and ag production groups. What is there lighter loss from these sources? Does it have to do with the reputation of their products? The economies of scale of the organizations?
Research Limitations:
What are the limitations of the econometric model? More information is needed here. The authors still aggregate prices. Moreover, they don’t have distance traveled, which can result in omitted variable bias. Using the average auction price is a major limitation as the auction price of the particular fruit or vegetable should influence the food loss for that specific food. While this point was improved, the authors still are aggregating. [Line 400]: Isn’t the assessment of the evaluation team subjective to some degree? Why are the authors making an objectivity claim?
Table 1:
Authors may include statistically significant symbols (i.e. *, **, ***) to show the results of two-sample t-tests between fruit and vegetable datasets. At the same time, the authors do speak about differences between food losses between fruits and vegetables. The authors should at least look at the differences between food loss weights.Table 2:
Why are the authors not including the market price_fruit and market price_vegetable in Colum A instead of market price_all? The authors shouldn’t aggregate the price for this specification. Are the authors reporting adjusted R^2s? Moreover, the authors should comment about the low R^2 levels in the models. The authors responded that low R^2 are common in large datasets. Provide a source that backs this statement.Table 3
Why didn’t the authors present a table similar to Table 3 for fruits?Table 4
Table 4 should be analogous to Table 2. Do not aggregate pricesAuthor Response
Please see the attchment
